# Patient-reported physical functioning is limited in almost half of critical illness survivors 1-year after ICU-admission: A retrospective single-centre study

Lise F. E. Beumeler[1¤]*, Anja van Wieren[2], Hanneke Buter[2], Tim van Zutphen[1], Nynke A. Bruins[2], Corine M. de Jager[2], Matty Koopmans[2], Gerjan J. Navis[3], E. Christiaan Boerma[2]

1 Campus Fryslân, University of Groningen, Leeuwarden, The Netherlands, 2 Department of Intensive Care, Medical Centre Leeuwarden, Leeuwarden, The Netherlands, 3 Faculty of Medical Sciences, University Medical Centre Groningen, Groningen, The Netherlands

¤ Current address: Campus Fryslân, University of Groningen, Leeuwarden, The Netherlands
* lisebeumeler@gmail.com

**Data Availability Statement:** All data are available from the Zenodo database (http://doi.org/10.5281/zenodo.4091035).

## Abstract

Post-intensive care unit (ICU) sequelae, including physical and mental health problems, are relatively unexplored. Characteristics commonly used to predict outcome lack prognostic value when it comes to long-term physical recovery. Therefore, the objective of this study was to assess the incidence of non-recovery in long-stay ICU-patients. In this single-centre study, retrospective data of adults with an ICU stay >48 hours who visited the specialized post-ICU clinic, and completed the Dutch RAND 36-item Short Form questionnaire at 3 and 12 months post-ICU, were retrieved from electronic patient records. In cases where physical functioning scores at 12 months were below reference values, patients were allocated to the physical non-recovery (NR) group. Significantly different baseline and (post-)ICU-characteristics were assessed for correlations with physical recovery at 12 months post-ICU. Of 250 patients, 110 (44%) fulfilled the criteria for the NR-group. Neither the severity of illness, type of admission, nor presence of sepsis did not differ between groups. However, NR-patients had a higher age, were more often female, and had a higher incidence of co-morbidities. Shorter LOS ICU, lower incidence of medical comorbidities, and better physical performance at 3 months were significantly correlated with 1-year physical recovery. Comorbidities and reduced physical functioning at 3 months were identified as independent risk-factors for long-term physical non-recovery. In conclusion, a substantial proportion of long-stay ICU-patients who visited the standard care post-ICU clinic did not fulfil the criteria for full physical recovery at 12 months post-ICU. Commonly used ICU-characteristics, such as severity of illness, do not have sufficient prognostic value when it comes to long-term recovery of health-related quality of life.

**Funding:** The author(s) received no specific funding for this work.

**Competing interests:** The authors have declared that no competing interests exist.

## Introduction

Over the past decades, both hospital mortality and long-term mortality of intensive care unit (ICU) patients have declined [1, 2]. However, post-ICU recovery has been shown to be complex and eminently heterogeneous. Clinical evaluations of recovery after ICU-admission commonly focus on mortality and on the ability to perform basic activities of daily living (ADL). Indeed, ADL-performance seems to be a strong indicator of self-efficacy at the moment of ICU or hospital discharge and is often used to identify short-term care needs [3]. However, these outcomes may be poor estimates of long-term prognosis. Thus, this short-term focus on recovery often does not equate the impact of critical illness in the long run.

Long-term recovery after critical illness is often characterised by a range of physical and mental impairments, e.g. ICU-acquired weakness, cognitive decline, and emotional distress, among others [4, 5]. It is commonly known that persistence of impaired physical health interferes with successful rehabilitation, which is an important issue for ICU-patients and their family. Pre-ICU physical health, demographic factors and disease aetiology influence physical recovery to some extent [6]. Still, ICU-admission and the highly specialized treatment that comes with it seem to play an equal role in the aetiology of long-term non-recovery. The finding that patients with a prolonged length of stay (LOS) are specifically burdened with long-term health problems illustrates the importance of this major life event as a separate risk factor for non-recovery [7].

To fill this deficiency in prognostic power, recent developments have been focused on health-related quality of life (HRQoL) as an indicator for long term recovery [8]. HRQoL-measurements encompass an integrative approach to physical, mental and social health of ICU-survivors. More specifically, both reduced physical health and being (partially) disabled have been shown to directly reflect on long-term HRQoL after ICU-discharge [9, 10]. To develop interventions which would improve HRQoL, specific target groups and risk factors with prognostic value regarding long-term non-recovery need to be determined. However, previous literature has often reported heterogenic results and were characterized by an overrepresentation of elective surgical ICU-patients, due to the ability to perform baseline tests [11]. Furthermore, loss to follow-up is a serious limitation concerning the implications of results [12]. These disadvantages make the generalization of previously obtained findings troublesome.

In conclusion, characteristics commonly used to predict outcome in ICU-patients lack sufficient prognostic value when it comes to long-term physical recovery. The paucity of predictors makes the identification of subgroups which fail to recover problematic. An integrative assessment of HRQoL has the potential to unravel the complexity of non-recovery in ICU-survivors. In this study we aimed to assess the patient-reported incidence of non-recovery in long-stay ICU-patients. Furthermore, factors in relation to various points in time (from baseline to 3 months post ICU-discharge) were considered for long-term (1 year) non-recovery.

## Materials and methods

### Study population

This retrospective, single-centre study was performed in a tertiary teaching hospital with a closed-format ICU. Local protocol dictates that all ICU-patients with an LOS ICU ≥48 hours receive an invitation for the specialized post-ICU clinic at 3 months after ICU-discharge. In line with this protocol, patients fill in an HRQoL-questionnaire and return it before their visit. At 12 months, patients repeat the questionnaire and return by mail. Data from all patients admitted to the ICU between 2012 and 2018 who completed and returned the questionnaire

were retrieved for this study. Data of patients who did not survive until the 1 year follow-up, were lost to follow-up, or who did not complete the physical functioning (PF) domain of the questionnaire were excluded from analysis (Fig 1).

## Ethical considerations

Due to the retrospective nature of this study, a local ethical committee determined this study was eligible to be assessed as a nWMO-research project (Regionale Toetsingscommissie Patiëntgebonden Onderzoek, Leeuwarden, The Netherlands; nWMO-number: nWMO 358). The need for informed consent was waived by the ethical committee as the patient-record data was analyzed anonymously.

## Data collection

In the specialized post-ICU clinic, the Dutch translation of the RAND-36 item Health Survey (RAND-36), which is very similar to the Medical Outcome Study Short-Form-36 (MOS SF-36), was used to evaluate HRQoL [13]. This questionnaire consisted of nine subscales used to assess physical functioning, social functioning, role limitations due to physical or emotional problems, mental health, energy/fatigue, pain, general health perception, and experienced changes in health during the past four weeks. In each subscale, a score was given from 0 to 100, in which higher scores represented better HRQoL. Effectiveness, validity and reliability of the RAND-36 have been tested repeatedly in both the general population and numerous patient groups, including ICU-patients [14].

According to standard protocol, PF-domain scores of the RAND-36 were collected at 3 and 12 months after ICU-discharge. These scores were compared to a reference value of healthy individuals aged 65 to 75. A margin was taken into account, resulting in a reference value PF-

**Fig 1. Flowchart of retrieved study data.**

domain score of 65 [13]. All patients with a PF-domain score below 65 were considered as part of the non-recovery (NR)-group [median score: 35 [20–50]). Simultaneously, patients with a PF-domain score equal or above 65 were assigned to the recovery (R)-group (median score: 85 [75–95]). Baseline characteristics and usual care data were collected from electronic patient files. Three patients did not complete all questions of the PF-domain of the RAND-36 and were subsequently excluded from analysis.

Standard care data on physical functioning were retrieved from electronic patient files (for overview, see Table 1). Functional status, muscle strength, walking distance, mobility, and balance were taken into account. Functional status, i.e. ADL-performance, at discharge and 3 months after ICU-admission, was assessed using the Barthel Index Score (BIS), a 10-item survey with a score range of 0–20 points [15]. Hand grip strength of the right hand (kg), also at discharge and 3 months after ICU-admission, was measured using a handheld dynamometer and transformed to a score relative to an age-adjusted reference value [16]. Mobility was tested using the Morton Mobility Index (MMI) score at discharge and 3 months after ICU-admission [17]. This survey consisted of 15 items with a score range of 0–19 points which were transformed to a relative score from 0–100 percent. Additionally, walking distance and balance were measured at the specialized post-ICU clinic at 3 months after ICU-discharge using the 14-item Berg Balance Scale (BBS), with a score range of 0–56 and the 6-minute walking test (6-MWT), which measured the distance (m) walked in 6 minutes relative to an age-adjusted reference value (%) [18, 19]. In all physical tests a higher score indicated a better outcome. After retrieval, all data were coded and processed in accordance with the General Data Protection Regulation.

## Statistical analysis

All data were extracted from the electronic patient files and transferred to a coded data file in January, 2019. Variables were summarised as median [interquartile range, IQR] and number (percentage) for continuous or categorical variables, respectively. At baseline, ICU-discharge, and 3 months after ICU-discharge, characteristics were compared between the R-group and NR-group using either the Mann-Whitney U test or the 2-sided Pearson Chi-Square test, as appropriate. Variables with a clinically relevant difference ($p \leq 0.25$) were considered for further analysis. A sample size-appropriate selection of these variables was made using both clinical experience and relevant literature regarding their already known effect on recovery and outcome. This selection was assessed for significant correlations with recovery, i.e. reaching the age-adjusted reference value. Spearman's rank correlation coefficient (rho) was computed and tested for significance. Finally, a binomial regression analysis was used to identify independent risk factors at ICU-discharge for non-recovery. Throughout the analysis, a 2-sided p-value of 0.05 was considered statistically significant. All statistical analyses were conducted using SPSS Statistics for Windows, Version 24.0 (IBM, Chicago, IL, USA).

**Table 1. Timeline of standard care data collection.**

| | |
|---|---|
| **ICU-admission** | Sex, Age, APACHE III-score, Comorbidities, Admission type, Diagnosis 'Sepsis', Diagnosis 'After CPR' |
| **ICU-discharge** | LOS ICU, Days of mechanical ventilation, Need for renal replacement therapy, Barthel Index Score, Hand grip strength (right hand), Morton Mobility Index score |
| **3 month visit post-ICU outpatient clinic** | Barthel Index Score, Hand grip strength, Morton Mobility Index score, 6-Minute Walking Test score, Berg Balance Scale, RAND-36 physical functioning subscale score |
| **12 month follow up** | RAND-36 physical functioning subscale score |

Before conducting a multivariate analysis, data were inspected for missing values. The main reason for missing data was related to the delayed implementation of physical functioning measurement in the ICU and in the specialised post-ICU clinic. Variables with ten percent or more missing values were supplemented using multiple imputations (S1 Table). The assumption was made that the majority of the missing data was missing completely at random, as this was mainly caused by a delayed implementation of the physical measurements. Five imputed datasets were created using predictive mean matching in order to correct for non-normality. Binomial regression analysis was applied using the backward elimination enter method.

## Results

### Identification of R-group and NR-group

A total of 250 patients visited the standard care post-ICU clinic at 3 months after ICU-discharge and completed the PF-domain of the RAND-36 questionnaire at 3 and 12 months. PF-domain scores at 12 months were compared to age-adjusted reference values [13]; median age was 67 [58–74]. Concurrently, 110 patients (44%) were assigned to the NR-group (PF-subscale score at 12 months: 35 [20–50]) and 140 (56%) to the R-group (PF-subscale score at 12 months: 85 [75–95]). In a univariate analysis, NR-patients had a significantly lower PF-domain score at 3 months compared to the R-patients (PF-subscale score at 3 months: 38 [25–55], 75 [60–85], respectively. $p < 0.001$). Additionally, PF did not change significantly over time in the NR-group. Contrarily, PF-domain scores of R-patients were higher at 3 months after discharge and increased significantly over time ($p < 0.001$) (Fig 2).

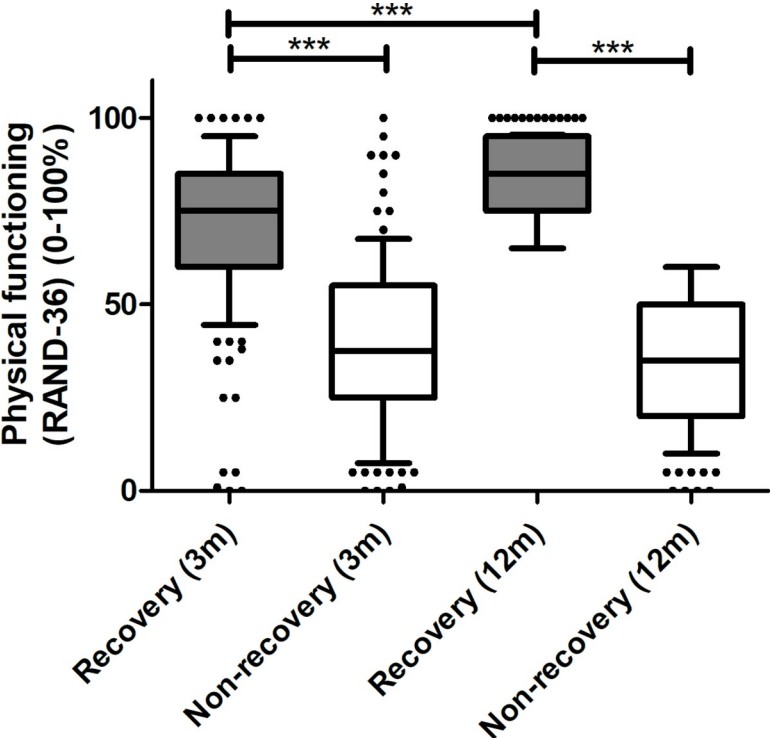

**Fig 2. Median and 10-90th percentile of physical functioning domain score (RAND-36) of R- and NR-patients at 3 and 12 months after ICU-admission.** *** $p < 0.001$.

## Baseline: & ICU-characteristics

Further univariate comparison of baseline characteristics comparing NR- to R-patients revealed significant differences in demographic factors, comorbidities, disease aetiology, and ICU-morbidity. NR-patients had a higher age (p <0.05), were more often of the female sex (p < 0.05), and had a higher incidence of co-morbidities (i.e. presence of pre-existing comorbidities; p < 0.001). Furthermore, NR-patients had a longer LOS ICU, more days on mechanical ventilation, and a higher incidence of renal replacement therapy (p <0.05). However, patients in the R-group there was a higher incidence of cardiopulmonary resuscitation as primary reason for ICU-admission (p < 0.05). There were no differences in severity of illness score (APACHE III), admission type, and presence of sepsis (Table 2A).

## Physical functioning at discharge and 3 months after ICU-admission

NR-patients performed worse on physical functioning compared to R-patients at ICU-discharge with regards to functional status (p < 0.01) and muscle strength (p < 0.01). These

**Table 2. A. Baseline—& ICU-characteristics, B. 3-months post-ICU admission measurements.**

| Characteristics | All, n = 250 | NR, n = 110 (44.0%) | R, n = 140 (56.0%) | *p*-value |
|---|---|---|---|---|
| **A.** | | | | |
| **Demographic factors** | | | | |
| *Female, n (%)* | 94 (37.6) | 49 (44.5) | 45 (32.1) | .044 * |
| *Age* | 67 [58–74] | 68 [60–76] | 66 [57–72] | .016 * |
| *APACHE III* | 79 [59–98] | 76 [60–98] | 82 [56–99] | .565 |
| **Comorbidities** | | | | |
| *Comorbidities, n (%)* | 114 (45.6) | 67 (60.9) | 47 (33.6) | < .001 *** |
| **Aetiology** | | | | |
| *Admission, n (%)* | | | | |
| Medical | 131 (52.4) | 55 (50.0) | 76 (54.3) | .433 |
| Elective surgical | 43 (17.2) | 17 (15.5) | 26 (18.6) | |
| Acute surgical | 76 (30.4) | 38 (34.5) | 38 (27.1) | |
| *Sepsis, n (%)* | 82 (32.8) | 40 (36.4) | 42 (30.0) | .287 |
| *CPR, n (%)* | 49 (19.6) | 15 (13.6) | 34 (24.3) | .035 * |
| **ICU morbidity** | | | | |
| *LOS ICU* | 11 [6–20] | 12 [7–26] | 10 [5–18] | .011 * |
| *Mechanical ventilation (days)* | 6 [3–12] | 7 [4–19] | 5 [3–9] | .035 * |
| *Renal replacement therapy (CVVH)* | 50 (20.0) | 30 (27.3) | 20 (14.3) | .010 * |
| **B.** | | | | |
| **PF, at ICU-discharge** | | | | |
| *BIS (0–20)* | 11 [6–16] | 8 [5–13] | 14 [8–17] | .001 ** |
| *Hand grip strength right (%)* | 57.30 [40.00–85.26] | 48.99 [31.60–69.28] | 68.16 [47.31–92.10] | .006 ** |
| *MMI (0–100)* | 30 [20–44] | 30 [20–42] | 30 [29–45] | .128 |
| **PF, 3 months after ICU-admission** | | | | |
| *BIS (0–20)* | 20 [20–20] | 20 [18–20] | 20 [20–20] | < .001 *** |
| *MMI (0–100)* | 85 [74–85] | 74 [62–85] | 85 [85–100] | < .001 *** |
| *6-MWT (%)* | 86.00 [68.75–97.00] | 76.00 [62.00–88.00] | 92.00 [80.00–103.00] | < .001 *** |
| *BBS (0–56)* | 53 [49–56] | 49 [44–54] | 55 [52–56] | < .001 *** |
| *Hand grip strength right (%)* | 96.15 [81.15–116.80] | 89.59 [70.70–105.00] | 100.89 [86.81–122.12] | < .001 *** |

*Abbreviations*: APACHE, Acute Physiology and Chronic Health Evaluation; CPR, cardiopulmonary resuscitation; LOS ICU, Length of stay Intensive Care Unit; CVVH, Continuous Veno-Venous Hemofiltration; BIS, Barthel index score; MMI, Morton Mobility Index; 6-MWT, 6-minute walk test; BBS, Berg balance scale.

differences persisted 3 months after ICU-admission (p < 0.001). Furthermore, NR-patients performed worse on walking distance, mobility and balance tests compared to R-patients 3 months after ICU-admission (p < 0.001). No significant difference in mobility was found between these patient groups at ICU-discharge (Table 2B).

## Correlations with recovery

Spearman's rank order correlation was computed in order to assess the correlation of recovery measures with those upon baseline-, ICU-discharge- and 33 month post-ICU-measures, i.e. reaching the age-adjusted reference value of the PF-domain score at 12 months after ICU-admission (Fig 3). The incidence of medical comorbidities at ICU-admission was weakly negatively correlated with recovery ($r_s$ = -.27, p < .001). Of the measures taken at ICU-discharge, LOS ICU, physical performance (i.e. BIS), and hand grip strength were weakly positively correlated with recovery ($r_s$ = .06, p < .05, $r_s$ = .25, p < .001, $r_s$ = .29, p < .01, respectively). Physical performance and hand grip strength at 3 months after discharge were also weakly positively correlated with recovery ($r_s$ = .33, $r_s$ = .27, p < .001). Additionally, walking distance (6-MWT), mobility (MMI), balance (BBS), and PF-domain score at 3 months after ICU-admission were moderately positively correlated with recovery ($r_s$ = .40, $r_s$ = .37, $r_s$ = .43, $r_s$ = .60, p < .001, respectively). It is of note that there was no significant correlation of either baseline APACHE III or mobility scores at ICU-discharge with recovery.

## Independent risk-factors

Finally, the independent predictive value of commonly used baseline characteristics and measures for physical functioning at 3 months after ICU-admission was assessed. A binary logistic regression was performed to ascertain the effects of LOS ICU, the APACHE III score, the incidence of medical comorbidities at baseline, as well as walking distance, mobility, balance, hand grip strength and physical functioning at 3 months after ICU-admission on the likelihood of recovery of physical functioning at 12 months after ICU-admission. Patients with pre-existing comorbidities may be less likely to improve in physical functioning after 12 months (original data: OR .620, CI .279–1.377; p = .241, corrected for missing values: OR .380 CI .197-.734; p = .004). A higher balance testing score at 3 months after admission was associated with an

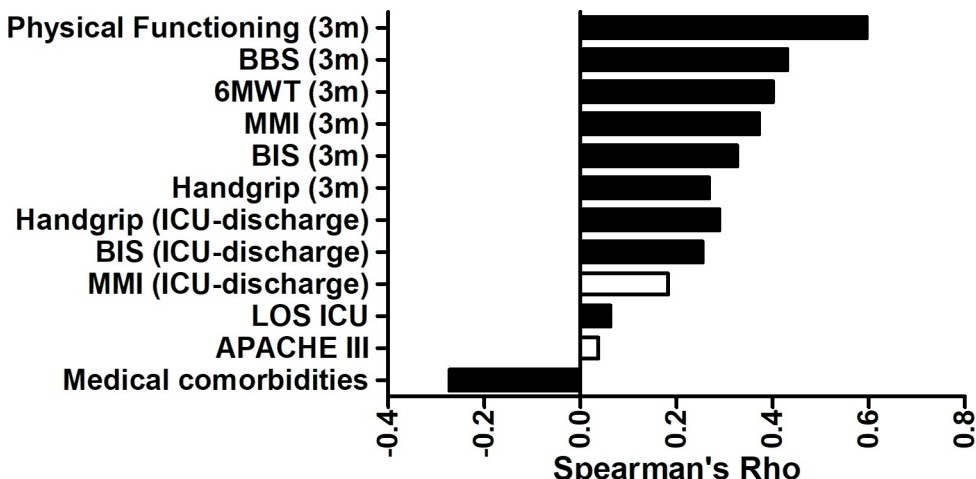

**Fig 3. Pairwise correlations with recovery at 12 months after ICU-admission.** Significant correlations visualised in black.

increased chance of recovery of physical functioning (original data: OR 1.105 CI 1.025- .192; p = .009, corrected for missing values: OR 1.066 CI 1.005–1.132; p = .034). Finally, a higher RAND-36 physical functioning subscale score at 3 months after ICU-admission increases the likelihood of recovery at 12 months (original data: OR 1.058 CI 1.037–1.080; p < .001, corrected for missing values: OR 1.052 CI 1.036–1.069; p < .001). The logistic regression model was statistically significant (original data: $\chi 2(3) = 79.1$, p < .001). The model explained 50.0% (Nagelkerke $R^2$ original data) of the variance in recovery of physical functioning and correctly classified 78.2% of cases.

## Discussion

In this retrospective study, a substantial proportion of patients (44%) who visited the standard care post-ICU clinic did not fulfil criteria for full physical recovery at 12 months after ICU-admission. We identified impaired physical functioning at 3 months (assessed as decreased balance and RAND-36 physical functioning subscale scores), as well as the presence of comorbidities at baseline as independent risk factors associated with non-recovery at 12 months. Thus, prediction of further (non-)recovery can only be made with reasonable accuracy 3 months after admission. In contrast, LOS ICU, duration of mechanical ventilation, and renal replacement therapy lost significance in our multivariate model. Moreover, well-established risk factors for hospital mortality, including APACHE III, as well as medical admission type and presence of sepsis, were equal in patients with and without full physical recovery at 12 months.

The observed percentage of patients in de NR-group was in line with previous literature. In general, comparison with the existing literature is hampered by differences in HRQoL-scales used, a focus on specific subgroups, and limited follow-up time in small study populations [20]. In a comparable Dutch general long-stay ICU population, SF-36-derived functional status one year post-ICU was limited in 54 percent of patients, as compared to an age-adjusted reference population [7]. Even in a substantially younger Norwegian mixed-ICU population, the majority of patients did not return to the age-adjusted physical functioning SF-36-score at one year post-ICU [21]. In a large cohort of long-stay ICU-patients with sepsis as the primary reason for ICU-admission, the mean SF-36 physical functioning at 12 months was 41 ± 35 percent [22]. Furthermore, in a small subset of acute respiratory distress syndrome (ARDS) survivors, 43 percent had self-reported functional limitations 12 months after ICU-admission [23]. The overall picture that emerges from these data is that a substantial amount of patients do not recover to the fullest extent.

Our data confirmed the presence of comorbidities as an independent risk-factor for the absence of full physical recovery, as reported by others [24]. However, in contrast to the general perception that severity of illness score and age are risk-factors for reduced HRQoL as well, the APACHE III score and age were not independently associated in our multivariate model [25]. In a general ICU-population with a median APACHE II score of 11, HRQoL after one year was not different either between patients with high and low Sequential Organ Failure Assessment (SOFA) scores [26]. In contrast, in a large cohort of patients with pneumonia and/ or sepsis, the Simplified Acute Physiology Score (SAPS) II was an independent predictor for reduced HRQoL. However, this effect was completely attributable to the difference between Q1 (lowest SAPS-score) and the rest of the group, whereas there was no significant difference between Q2 to Q4. Furthermore, the presence of sepsis and age were not independently associated with HRQoL in this study, which is similar to the results in the present study [27]. In a recent retrospective study, several multivariate models were constructed to assess influential factors on one-year HRQoL, expressed as the EQ-5D derived utility index. Although the APACHE II and SOFA score, as well as age, were independently associated, the authors

underlined that the models at best explained 20–40 percent of the variability in utility index score [6]. The limitations of the available predictive models for outcome are generally well-recognised, since they are designed to predict hospital mortality instead of long-term outcome and HRQoL [28].

The most important distinction between previous models and our approach was related to the inclusion of HRQoL and physical status data at 3 months, which was done in order to predict HRQoL one year post-ICU. At first glance, it might be confusing to incorporate baseline characteristics from the moment of ICU-admission as well as post-ICU parameters. However, the results of the present study justify the notion that the net result of long-term outcome is not only determined by the severity of illness but also by the individual response of the patient to the insult. According to the Nagelkerke $R^2$, not only was our model able to explain 50 percent of the variability in physical functioning recovery, it also rendered the majority of baseline characteristics statistically insignificant. An additional explanation for the substantial goodness-of-fit lies in the bivariate distinction between patients who fully recover versus those who do not. This division seems to closely match the diversity in HRQoL-scores which occurs during recovery after critical illness.

Evidently, future studies incorporating physical functioning at 3 months after ICU-discharge are urgently needed. In addition, it is necessary to identify factors and sub-groups not only at 12 and 3 months but also at ICU-discharge, in order to potentially facilitate multidisciplinary aftercare programs which aim to improve long-term HRQoL as early as possible.

## Limitations

Even though these findings may contribute to a further understanding of the large diversity in recovery after critical illness, there were several limitations to our study to keep in mind. First of all, the retrospective, single centre design clearly limited the potential to generalise results. The patient population per hospital can be highly diverse, especially in critical care. With regards to our physical functioning data, there was a high amount of missing data due to loss to follow up. Furthermore, our study design lacked baseline measurements of HRQoL, identical to other studies regarding acute ICU patients. Henceforth, methods in order to resemble baseline measurements should be considered, such as proxy measurements [29]. Additionally, to take into account the pre-admission frailty of patients admitted to the ICU-ward would strengthen results.

Additionally, this study reports on the physical recovery of patients who were willing and/or are able to visit the post-ICU outpatient clinic at 3 and complete the questionnaires at 12 months after ICU-discharge. Consequently, the recovery of patients who were unable to conduct the previous is unknown. It should be noted that most patients lost to follow-up indicated an overwhelming amount of appointments with health-care professionals as the main reason to decline the post-ICU clinic invitation, rather than inability to attend due to physical or mental health problems. Future studies might be able to limit the number of lost to follow-up measurements by conducting proxy or measurements via telephone. Additionally, due to the retrospective nature of this study and the limited financial resources of the standard care post-ICU clinic, no in-person physical tests were conducted at 12 months after discharge. This would have improved the robustness of the data and should be taken into account in future studies regarding this subject.

## Conclusions

A significant amount of ICU-patients do not recover in physical functioning 1 year after ICU-admission. Pre-existing comorbidities and limited physical functioning at 3 months after ICU-

admission predict such a lack of long-term physical recovery. Future studies are needed to define subgroups of ICU-patients in need of a timely, multidisciplinary and personalised after-care program in order to improve long-term health-related quality of life in critical illness survivors.

## Supporting information

**S1 Table. Percentage of missing values and medians of the original and imputed dataset.** (DOCX)

## Author Contributions

**Conceptualization:** Lise F. E. Beumeler, Anja van Wieren, Hanneke Buter, Tim van Zutphen, Nynke A. Bruins, Corine M. de Jager, Matty Koopmans, Gerjan J. Navis, E. Christiaan Boerma.

**Data curation:** Lise F. E. Beumeler, Anja van Wieren, Nynke A. Bruins, Corine M. de Jager, Matty Koopmans.

**Formal analysis:** Lise F. E. Beumeler, Hanneke Buter, Matty Koopmans, E. Christiaan Boerma.

**Investigation:** Lise F. E. Beumeler, Hanneke Buter, Tim van Zutphen, E. Christiaan Boerma.

**Methodology:** Lise F. E. Beumeler, Hanneke Buter, Tim van Zutphen, Nynke A. Bruins, Matty Koopmans, Gerjan J. Navis, E. Christiaan Boerma.

**Project administration:** Lise F. E. Beumeler, Anja van Wieren, Nynke A. Bruins, Corine M. de Jager, Matty Koopmans, Gerjan J. Navis, E. Christiaan Boerma.

**Resources:** Lise F. E. Beumeler.

**Supervision:** Hanneke Buter, Tim van Zutphen, Matty Koopmans, Gerjan J. Navis, E. Christiaan Boerma.

**Validation:** Lise F. E. Beumeler, Hanneke Buter, Tim van Zutphen, Corine M. de Jager, E. Christiaan Boerma.

**Visualization:** Lise F. E. Beumeler, E. Christiaan Boerma.

**Writing – original draft:** Lise F. E. Beumeler, E. Christiaan Boerma.

**Writing – review & editing:** Lise F. E. Beumeler, Anja van Wieren, Hanneke Buter, Tim van Zutphen, Nynke A. Bruins, Corine M. de Jager, Matty Koopmans, Gerjan J. Navis, E. Christiaan Boerma.

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
