## [Decision Letter · Decision Letter 0]

7 Sep 2020

PONE-D-20-21277

Patient-reported physical functioning is limited in almost half of critical illness survivors 1-year after ICU-admission: a retrospective single-center study

PLOS ONE

Dear Dr. Beumeler,

Thank you for submitting your manuscript to PLOS ONE. After careful consideration, we feel that it has merit but does not fully meet PLOS ONE’s publication criteria as it currently stands. Therefore, we invite you to submit a revised version of the manuscript that addresses the points raised during the review process.

The reviewers have raised several metholdologic, and at least one research ethics concern in their critiques.  These must be addressed in a MAJOR REVISION of the manuscript to be further considered for publication.  

We look forward to receiving your revised manuscript.

Kind regards,

Scott Brakenridge, M.D.

Academic Editor

PLOS ONE

Additional Editor Comments:

There multiple methodological and at least one research ethics concern pointed out in the reviewers' critiques. These will need to be specifically, clearly and thoroughly addressed in a MAJOR REVISION of the manuscript, to be further considered for potential publication.

Journal Requirements:

2. In your Methods section please include the dates upon which authors accessed the clinical data sources used in this study.

4. Your ethics statement must appear in the Methods section of your manuscript. If your ethics statement is written in any section besides the Methods, please move it to the Methods section and delete it from any other section. Please also ensure that your ethics statement is included in your manuscript, as the ethics section of your online submission will not be published alongside your manuscript.

Reviewers' comments:

Reviewer's Responses to Questions

**Comments to the Author**

1. Is the manuscript technically sound, and do the data support the conclusions?

Reviewer #1: Partly

Reviewer #2: Yes

Reviewer #3: Yes

2. Has the statistical analysis been performed appropriately and rigorously? 

Reviewer #1: Yes

Reviewer #2: Yes

Reviewer #3: Yes

3. Have the authors made all data underlying the findings in their manuscript fully available?

Reviewer #1: Yes

Reviewer #2: Yes

Reviewer #3: Yes

4. Is the manuscript presented in an intelligible fashion and written in standard English?

Reviewer #1: Yes

Reviewer #2: No

Reviewer #3: No

5. Review Comments to the Author

Reviewer #1: I applaud the authors for investigating a noteworthy issue that, up until recent, has received little scientific attention. The overall findings are not surprising given the results of other studies assessing similar long-term outcomes among ICU patients. While the study is credible, it lacks the methodological prowess to create results that build upon existing literature.

I am unsure why objective, in-person physical function data was collected at discharge and 3 months, but not at 12 months. The robustness of the data would have been increased if the authors hadn’t relied solely on self-report at 12 months.

Also, the sample size would have been improved if some sort of proxy measure was taken to reflect physical function and recovery of those unable to return to the clinic. For instance, the SF-36 could have been completed over the phone.

Did the authors have any contact with the participants between the 3 and 12 month period? If so, this should be outlined in the the methods. If not, perhaps this would have helped decrease loss to follow-up over the course of the study.

The way the authors have the methods written makes it difficult to keep track of what measures were collected when. A diagram illustrating some sort of timeline (ie admission, discharge, 3-months, 12-months) along with the measures collected at each time point would be helpful.

Again, the overall aim of this study is worthwhile; however, the study holds substantial methodological limitations that make it worthy of publication.

Reviewer #2: In their manuscript “Patient-reported physical functioning is limited in almost half of critical illness survivors 1-year after ICU-admission: a retrospective single-center study” the authors report on recovery/non-recovery of ICU survivors and explore the role of potential risk factors. The data presented are interesting and can advance knowledge in the area. However, some points need to be further addressed.

1. The title of the study does not sound to be in line with aim (lines 101-102, to identify risk factors for long-term non-recovery at various time points from baseline to three months post-ICU discharge) and, in part, conclusions. The authors might consider rephrasing.

2. What was the hypothesis to contact this study?

3. The aim may better get rephrased to suggest that factors in relation to various time points (from baseline to three months post ICU-discharge) were considered for long-term (1 year) non-recovery.

4. Line 107. ‘were invited’ may be more appropriate than ‘are invited’.

5. Lines 133-134. Pls consider defining NR and R-group.

6. Results, #3.1. Pls consider also reporting PF values at 12 months for both groups, as this is a major variable to report.

7. Results, #3.4. This section refers to correlations of various variables with PF-score at 12 months; however, it seems to be written as referring in between-group comparisons (already done in #3.2.). Pls consider rephrasing. Also, at lines 218, 222 the authors mention tendency but report significant p values. Pls clarify.

8. Figure 3. Nice figure to graphically represent r coefficients.

9. Results, #3.5. The authors might include a table to report values (OR, CI, p) for all variables employed in the regression model. Also, what is meant by “pooled results”? Are they results corrected for missing values?

10. Both last paragraph of discussion (lines 302-315) and #5 refer to study limitations. What is the reason for separating? Pls clarify.

11. Conclusions, lines 327-330. This phrase might be more appropriate to include in discussion than conclusions, as it refers to future studies.

12. The authors might consider including in discussion a comment on the necessity to identify factors and sub-groups not only at 3 months after but also at hospital discharge to potentially facilitate rehabilitation interventions as early as possible.

Reviewer #3: Review:

Patient-reported physical functioning is limited in almost half of critical illness survivors 1-year after ICU-admission: a retrospective single-center study

Corresponding Author: Lise Frieda Elisabeth Beumeler

General:

I thank the authors for the opportunity to read their manuscript “Patient-reported physical functioning is limited in almost half of critical illness survivors 1-year after ICU-admission: a retrospective single-center study”. The authors investigated the post-intensive care recovery with a survey assessed by patients (Dutch-RAND 36-item Short Form).

1. I am very concerned about the study design. What is “retrospective” in this study? Study design sound more like an observational study. Which leads to the question “How did the study get submitted to the ethics committee”. As a retrospective study? Which does not correspond to the real study design. It seems questionable that an ethic committee waived the need for informed consent of an observational study. Even, when patient-related data were used and study-related procedures were used (Hand Grip strength, 6-MWT and BBS at discharge from ICU AND 3 month post-discharge). Please provide the correspondence to your local ethic committee including the study protocol (translated in English).

As it can be assumed that the study was submitted to the ethics committee as "retrospective", which does not correspond to the real study design. Please clarify this.

6. PLOS authors have the option to publish the peer review history of their article (what does this mean?). If published, this will include your full peer review and any attached files.

Reviewer #1: No

Reviewer #2: **Yes: **Eleftherios Karatzanos PhD

Reviewer #3: No

---

## [Author Response · Author response to Decision Letter 0]

19 Oct 2020

Dear academic editor and reviewers,

The authors would like to thank the reviewers for their elaborate and helpful comments regarding our submitted manuscript. The manuscript has been edited to address their concerns.

Please find the revised manuscript and related documents enclosed in this resubmission. The authors thank you again for your careful consideration.

Academic editor’s response:

1. The authors thank the editor for providing information regarding PLOS ONE’s style requirements. The manuscript and all additional files are adjusted accordingly. 

2. The date upon which the authors accessed the clinical data sources in this study was added to the Methods section.

3. After careful revision of the data the authors agree with the editor to make the minimal anonymized data set necessary to replicate our study findings accessible through “Beumeler, L.F.E. (2020). Patient-reported physical functioning is limited in almost half of critical illness survivors 1-year after ICU-admission: a retrospective single-centre study [Data set]. Zenodo. http://doi.org/10.5281/zenodo.4091035”. We thank the editor in advance for updating our Data Availability statement on our behalf.

4. The duplicate of our ethics statement has been removed from the manuscript. According to the response of the editor, it now only appears in the Methods section of the manuscript.

5. The appropriate captions for our supporting information file is added at the end of our manuscript and the in-text citation is updated.

Comments to the Author

The authors thank the reviewers for their elaborate response to the questions regarding the manuscript and their useful feedback. To adhere to the information provided by the questions regarding the integrity and language of the manuscript, the procedural text is clarified and the whole manuscript is assessed by a professional English language expert.

Comments to reviewer 1

The authors thank reviewer 1 for acknowledging the importance of the investigation of this issue. We agree with reviewer 1 regarding the complimentary nature of our findings. There have indeed been similar findings in other studies regarding long-term outcomes among ICU patients. However, differences in patient characteristics, ICU treatment methods, and after care facilities make it relevant to investigate these trajectories within a Dutch tertiary teaching hospital, as opposed to for instance a academic hospital (in The Netherlands or abroad). Furthermore, tertiary teaching hospitals may even treat more patients and also less complex cases compared to an academic hospital, which supports the relevance of the above. In our opinion it therefore adds insights to build upon within the development of locally applicable, personalised after care programs.

We thank reviewer 1 for addressing the confusion related to the lack of in-person follow up in physical functioning data at 12 months. We agree that inclusion of these measurements would greatly improve the robustness of the data. Due to the retrospective nature of this study, we were unable to obtain these data thus far. To clarify this issue, we added this limitation of the study to our manuscript. In an upcoming prospective study, we include several in-person follow up visits within the first year after discharge to overcome this issue.

Secondly, we agree with reviewer 1 that proxy measures or telephonic interviews regarding the questionnaires could have improved the sample size by reducing the number of patients lost to follow ups. To clarify, the standard care outpatient clinic always requests the patient to complete and return the questionnaire even if they are unable or unwilling to visit the clinic. This study aims to contribute to improving standard care practices in the recovery phase. However, regular follow up visits are still not common practice. Due to the retrospective nature of this study we were unable to improve this. This also applies to the idea regarding more regular follow up contact between patients and the workers in the clinic. Reviewer 1’s suggestion will be added to the manuscript as a recommendation for future studies.

Thirdly, we value reviewer 1’s comment regarding the possible difficulty readers may experience regarding keeping track of what measures were collected when. To clarify this process, we have added a timeline-table along with the measures collected at each time point, as suggested. We hope this makes the associated text easier to read. 

Finally, we value the effort reviewer 1 has made to help us improve the manuscript and clarify essential matters regarding our study methodology. 

Comments to reviewer 2

The authors thank reviewer 2 for their elaborate feedback on our manuscript. Below the points mentioned by reviewer 2 will be discussed to clarify the questions that arose while reading through the manuscript.

1. We agree with reviewer 2 that the title does not completely encompass the aim of this study. As mentioned in the objectives, we investigated the incidence of non-recovery in long-stay ICU-patients and identified risk-factors for non-recovery. We see that this might lead to some confusion. Concordantly, we adjusted the manuscript to clarify this issue. 

2. This retrospective patient-data study was designed to be exploratory.. The purpose of the study was to explore the area of recovery in physical functioning after long-term ICU-admission in our specific patient-group more thoroughly in order to develop hypotheses that can be tested in our future research. 

3. We thank reviewer 2 for their suggestion regarding rephrasing the aim of our study. The suggestion is added to the manuscript.

4. The suggested rephrasing has been adopted in the manuscript.

5. In line 133-134 the NR and R-group are written out fully. Also, the physical functioning subscale score medians with IQR are added to suffice in the request made by reviewer 2.

6. Physical functioning subscale scores at 12 and 3 months are clarified in the 3.1 result section.

7. We agree with reviewer 2 regarding the need to rephrase the results of the analyses for correlations with recovery at 12 months. This paragraph has been rephrased to better display the nature of these results. 

8. We thank reviewer 2 for the compliment regarding Figure 3. 

9. Regarding the results in 3.5, reviewer 2 suggests to include a table reporting values of the multivariate analysis. Although we agree with the need to display results in an orderly fashion, we prefer to textually report these results. However, we do agree with the raised question about the ‘pooled results’. Using this terminology might be confusing to the reader. Indeed, they entail the results corrected for missing values. Consequently, this is adjusted in the manuscript.

10. Reviewer 2 correctly points out that the last paragraph of the discussion should be included within the limitations. This is corrected in the manuscript.

11. +12 Following reviewer 2’s suggestions regarding the conclusions of this study, a short paragraph is added in the discussion section.

Comments to reviewer 3

The authors thank reviewer 3 for their effort to carefully review our manuscript. Additionally, we appreciate that reviewer 3 openly and clearly shares their concerns about the ethical considerations of this study. Following the comments, it is clear to us that the explanation of the study design is articulate enough. To make sure other readers do not have a similar reaction to this manuscript, several adjustments have been made to clarify the retrospective nature of this study. We guarantee this study is retrospective and think a more extensive explanation can mitigate reviewer 3’s concerns. All data used in this study was collected within our standard care after care clinic, by our ICU-nurses. This data was collected to facilitate proper after care, as the nurses and intensivists use the results in their consult with the patient and the correspondence that is send to the general practitioner and, in case of referral, to a physical therapist/psychologist/rehabilitation specialist. For instance, the hand grip strength, walking test score, and balance and mobility results are communicated to the physical therapist to give an idea of the current physical status of the patient. These data were all recorded within our patient-data-system. Before the start of this study, a study protocol was send to our local medical ethical committee (Regionale Toetsingscommissie Patiëntgebonden Onderzoek, Leeuwarden, The Netherlands; nWMO-number: nWMO 358). This ethical committee concluded our study protocol was indeed of a retrospective nature and therefore waived the need for informed consent. Both the researchers and the members of the ethical committee agreed it would be not possible and undesirable to obtain informed consent in retrospect. To further accommodate reviewer 3’s response, the English correspondence letter of our local ethical committee and the research protocol that was send for review are added in this resubmission.

We hope that our modifications render our manuscript in its current form suitable for publication in PLOS ONE.

Yours sincerely,

Lise Beumeler

---

## [Decision Letter · Decision Letter 1]

2 Dec 2020

Patient-reported physical functioning is limited in almost half of critical illness survivors 1-year after ICU-admission: a retrospective single-center study

PONE-D-20-21277R1

Dear Dr. Beumeler,

We’re pleased to inform you that your manuscript has been judged scientifically suitable for publication and will be formally accepted for publication once it meets all outstanding technical requirements.

Kind regards,

Scott Brakenridge, M.D.

Academic Editor

PLOS ONE

Additional Editor Comments (optional):

The editor believes the ethical concerns raised by the reviewers have been clarified and other major critiques regarding analyses and interpretation addressed.

Reviewers' comments:

Reviewer's Responses to Questions

**Comments to the Author**

1. If the authors have adequately addressed your comments raised in a previous round of review and you feel that this manuscript is now acceptable for publication, you may indicate that here to bypass the “Comments to the Author” section, enter your conflict of interest statement in the “Confidential to Editor” section, and submit your "Accept" recommendation.

Reviewer #2: All comments have been addressed

2. Is the manuscript technically sound, and do the data support the conclusions?

Reviewer #2: Yes

3. Has the statistical analysis been performed appropriately and rigorously? 

Reviewer #2: Yes

4. Have the authors made all data underlying the findings in their manuscript fully available?

Reviewer #2: Yes

5. Is the manuscript presented in an intelligible fashion and written in standard English?

Reviewer #2: Yes

6. Review Comments to the Author

Reviewer #2: Beumeler et al have improved the quality of their submitted manuscript.

All comments mentioned have been adequately addressed.

7. PLOS authors have the option to publish the peer review history of their article (what does this mean?). If published, this will include your full peer review and any attached files.

Reviewer #2: **Yes: **Eleftherios Karatzanos, PhD

---

## [Editor Report · Acceptance letter]

4 Dec 2020

PONE-D-20-21277R1 

Patient-reported physical functioning is limited in almost half of critical illness survivors 1-year after ICU-admission: a retrospective single-centre study 

Dear Dr. Beumeler:

I'm pleased to inform you that your manuscript has been deemed suitable for publication in PLOS ONE. Congratulations! Your manuscript is now with our production department. 

Kind regards, 

on behalf of

Dr. Scott Brakenridge 

Academic Editor

PLOS ONE